# Insight into Tar Formation Mechanism during Catalytic Pyrolysis of Biomass over Waste Aluminum Dross

Peng Liu [1,2], Li Liu [1], Zhengzhong Zhou [1,*], Haoran Yuan [1,3], Tao Zheng [3,4], Qigang Wu [1] and Huhe Taoli [1,2,*]

1   National-Local Joint Engineering Research Center of Biomass Refining and High-Quality Utilization, Institute of Urban and Rural Mining, Changzhou University, Changzhou 213164, China; liupeng@cczu.edu.cn (P.L.); 18000685@smail.cczu.edu.cn (L.L.); yuanhaoran@cczu.edu.cn (H.Y.); wqg1990@cczu.edu.cn (Q.W.)
2   Jiangsu Key Laboratory of Process Enhancement & New Energy Equipment Technology, Nanjing University of Technology, Nanjing 211816, China
3   Guangzhou Institute of Energy Conversion, Chinese Academy of Sciences, Guangzhou 510640, China; zhengtao@ms.giec.ac.cn
4   Innovation Academy for Green Manufacture, Chinese Academy of Sciences Beijing, Beijing 100190, China
*   Correspondence: zhouzhengzhong@cczu.edu.cn (Z.Z.); hhtaoli@cczu.edu.cn (H.T.); Tel.: +86-0519-85516132 (Z.Z. & H.T.)

**Abstract:** Tar is one of major products from biomass pyrolysis. Its formation mechanism in a catalytic pyrolysis system comprising pine sawdust and waste aluminum dross (AD) is investigated with the aid of analytical methods including thermogravimetric analysis (TG), Nuclear Magnetic Resonance (NMR), electron paramagnetic resonance (EPR), and gas chromatography coupling with mass spectrometry (GC-MS). The results show that AD plays a vital role in cleavage of C-O bonds to enhance selective formation of furans, ketones, and phenols. The catalytic pyrolysis is initiated by active C-O-M intermediate formation that accelerates C-O bond cleavage and generates great amounts of free radicals to 1020 spins/g at 300–500 °C. Compared with pure pine pyrolysis, the percentage of glucosidic bonds from cellulose decreases from 14.00% to 9.66% at 500 °C; the etherified guaiacyl is more actively ruptured and disappears at 700 °C. Furans and ketones increase from 17.45% to 22.23% and 6.71% to 10.80% at 500 °C, respectively. Phenols increase from 66.75% to 71.57%. The preferential production of higher value-added products via catalytic pyrolysis between biomass and industrial wastes may bring new insight to the simultaneous valorization of agricultural, municipal, and industrial waste.

**Keywords:** catalytic pyrolysis; tar formation; carbon structure; waste aluminum dross; free radical

## 1. Introduction

Biomass is the unique carbon-based renewable energy source and a potential supplement to fossil fuels in the worldwide energy system [1,2]. Biomass, composed of cellulose, hemicellulose, and lignin, can be converted into useful fuel gases or chemicals via gasification [3,4]. Pyrolysis is the first step during biomass gasification, which includes a set of complex reactions such as primary pyrolysis and secondary reactions of char forming, depolymerization, and fragmentation [5,6]. Tar is one of the primary products, taking part in the subsequent coking reaction and gasification reactions. The coking reaction forms tiny carbon particles, corroding the equipment and clogging the pipes during industrial applications [7,8]. Tar formation also decreases the conversion rate of biomass to fuel gas, impacting negatively the energy efficiency of a gasification system [9].

Tar is formed from free radical reactions during pyrolysis, which is influenced by the operating conditions and biomass compositions [10,11]. Researchers have studied the free radical reaction during biomass pyrolysis using model compounds to explain the formation mechanisms of tar, gas, and coke. Zheng [12] studied the initial reaction mechanisms of cellulose pyrolysis and revealed the detailed free radical reactions with the

aid of ReaxFF molecular dynamics, which agreed well with the cellulose pyrolysis–GC-MS data. Huang et al. [13] also investigated the reaction pathways of hemicellulose models by the density functional theory. With respect to free radical analysis during biomass pyrolysis, lignin or lignin model compounds are usually used to predict the characteristics of free radicals. Britt and co-workers [14–16] used β-O-4 dimeric phenolic compounds to propose a complex reaction pathway dominated by free radical reactions. Kim et al. [17] reported the pyrolysis of methoxy substituted α-O-4 dimeric phenolic compounds to investigate the effects of methoxy groups on lignin pyrolysis; free radicals were detected by electron paramagnetic resonance (EPR). Kibet et al. [18] used in-situ EPR to detect free radicals to resolve the difficulty of directly observing reaction intermediates from lignin pyrolysis. However, biomass pyrolysis behavior is more complex than the model compounds or lignin due to the interaction among cellulose, hemicellulose, and lignin. Wang and Liu [19,20] investigated the interaction of biomass components and its influence on pyrolysis. They found that pyrolysis characteristics of biomass cannot be predicted through summation of the individual components. In spite of a number of studies of the pyrolysis mechanism, tar formation dominated by free radical reactions is still not clearly understood. Pyrolysis on a catalyst may have an influence on free radical variation converting tar to fuel gas.

Active alumina is widely used as a catalytic support [21–24] for tar removal or a heat carrier in a fluidized catalytic cracking reactor [25], which embodies high efficiency for heat transfer and accelerates the water to gas reaction. Sun [26] showed the catalytic effect of composite alumina on the pyrolysis of sewage sludge in a fluidized bed reactor. The composition in aluminum dross (AD) is similar, containing alkali metal, Mg, and Fe, which may be used as catalyst for C-C and C-O bond activation [27–29] during biomass pyrolysis. However, the mechanism of the AD catalytic effect on biomass pyrolysis is rarely reported.

In order to expatiate the formation and removal of tar and the catalytic effect of AD, AD catalyzed biomass pyrolysis is conducted and the free radical reactions are scrutinized. To the best of our knowledge, this specific research area has received little attention and few works are published. This study compares the results of thermogravimetric analysis (TG) and $^{13}$C-NMR to elaborate the molecular structural evolution of an AD catalyzed biomass. Tar composition is analyzed by GC-MS. The mechanism and conversion pathway on biomass pyrolysis dominated by free radical reactions are systematically reported. These data may facilitate the process optimization and reactor design of commercial-scale biomass catalytic pyrolysis or gasification, and provide further guidance for innovating environmentally friendly methods of waste aluminum dross utilization.

## 2. Materials and Methods

### 2.1. Materials

Pine Sawdust

Pine sawdust was collected from the south of Anhui province in China, were used as the raw material. They were ground to <0.5 mm and stored under a cryogenic environment for subsequent analysis. AD was produced from the secondary smelters during the dross recycling process of a factory in Changzhou, which is identified as the hazardous wastes to be disposed of that is rich in $Al_2O_3$ as a catalyst for pyrolysis. The main chemical composition of AD analyzed by X-ray fluorescence spectrometer (XRF, S8 Tiger, Bruker AXS, Karlsruhe, Germany) was 78.38% of $Al_2O_3$, 7.77% of $SiO_2$, 5.18% of $Na_2O$, 4.51% of MgO, 0.68% of CuO, 0.74% of $Fe_2O_3$, 0.51% of $K_2O$, 0.39% CaO, and 0.19% of ZnO. They were ground to <0.1 mm for catalytic pyrolysis.

### 2.2. Methods

#### 2.2.1. Pyrolysis

Pyrolysis tests were carried out in a vertical quartz tubular reactor. The reactor was made of a quartz tube with an external diameter of 40 mm and an overall length of 750 mm. The gas–liquid separator was connected with the reactor through ground glass, which is shown in Figure 1. The control was 2.4 g of pine sawdust and the testing sample was a

mixture of 2.4 g pine sawdust and 0.6 g AD. $N_2$ was led into the reactor at 90 mL/min as a carrier gas. The reactor was heated to a set temperature at a heating rate of 10 °C/min. The experiment was repeated more than three times. The char from the pine pyrolysis was assigned as PC, and the composite from co-pyrolysis of AD and pine was assigned as PADC.

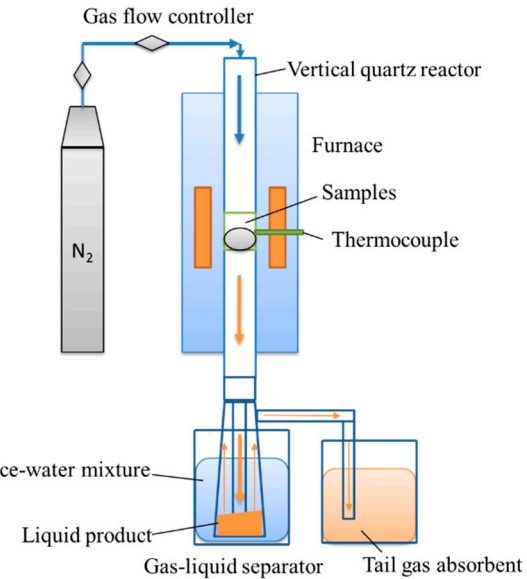

**Figure 1.** Diagram of the vertical fixed-bed reactor system.

### 2.2.2. Mass Change of Sample by TGA

The thermal decomposition of the biomass was carried out in a thermogravimetric analyzer (Pyris 1 TGA, Perkin Elmer, Waltham, MA, USA). In the TG experiment, 15 mg of sample was heated from ambient temperature to 750 °C at a heating rate of 10 °C/min, in an inert environment maintained by a 99.999% helium (He) stream flowing at 20 mL/min. The range of temperature lower than the point where the weight loss rate increased to 0.1%/°C was assigned as the initial pyrolysis temperature, and the range of temperature beyond the point where the weight loss rate decreased to 0.1%/°C was assigned as the terminal pyrolysis temperature. The TGA analysis was repeated more than three times. The temperature range of 100–550 °C was analyzed by the curve-fitting method in this study.

### 2.2.3. Composition Analysis of Pyrolysis Tar by GC-MS

The pyrolysis tar was dissolved in dichloromethane first. Its composition was then analyzed by gas chromatography (GC, Clarus 680, PerkinElmer, Waltham, MA, USA) coupled with a mass spectrometer (MS, Clarus SQ 8 T, PerkinElmer, Waltham, MA, USA) operating in the electron impact ionization mode using an Agilent DB-17MS column. The chemical workstation NIST14 (US National Institute of Standards and Technology) standard spectrum library and area normalization method was used for qualitative and quantitative analysis of the components. The GC-MS analysis was repeated more than three times.

### 2.2.4. Characterization of Carbon Structure and Free Radical Concentration of Solid Product from Pyrolysis

The carbon structure of solid products was investigated by [13]C-NMR. [13]C MAS NMR experiments of the biomass were performed on an Agilent 600 DD2 spectrometer at a resonance frequency of 150.15 MHz as described in our previous work [30,31]. The chemical shift assigned carbon structures are given in Table 1.

**Table 1.** Assignment of chemical shift.

| Chemical Shift/ppm | 178 | 164 | 146 | 129 | 116 | 109 | 101 |
|---|---|---|---|---|---|---|---|
| Assignment | Carbohydrate; -COO-R, CH3-COO- | Lignins; S [1]3(e [2]) | Lignins; G [3]1(e), G4(e), S3(ne [4]), S5(ne) | Lignins; G1(e), S1(ne), S4(ne) | Lignins; G6, G2; S6, S2 | Celluloses; C1 | Hemicelluloses; C1 |
| Chemical shift/ppm | 97 | 85 | 80 | 72 | 68 | 60 | 52 |
| Assignment | Hemicelluloses; C4 | Ordered celluloses; C4 | Amorphous celluloses; C4 lignin; C$\beta$ | Hemicelluloses and cellulose; C2,3,5,Lignins; C$\alpha$ | Celluloses; C6 | Lignins; C$\gamma$ | Lignins; OCH3 |
| Chemical shift/ppm | 43 | 30 | 15 | | | | |
| Assignment | Aliphatic C-C | Aliphatic CH2 | Hemicelluloses; CH3-COO- | | | | |

1 S is syringyl unit; 2, e is etherified; 3, G is guaiacyl units; 4, ne is nonetherified.

The free radical characteristics of samples were analyzed by EPR spectroscopy. The EPR experiments were carried out using an X-band in a Bruker EMX-8/2.7 spectrometer (Germany) at 25 °C. The amount of free radicals (N) was calculated using the Bruker computer software WinEPR Acquisition. The quantitative analysis of free radicals (N) was calibrated by a 2,2-diphenyl-1-picrylhydrazyl (DPPH) standard. The calculation method was described in the article published by Lv et al. [32].

## 3. Results and Discussion

### 3.1. Effect of Aluminum Dross on Free Radical Generation during Catalytic Pyrolysis

3.1.1. Effect of Aluminum Dross on DTG Behavior

The free radical reaction mechanism of biomass pyrolysis comprised radical generation by covalent bond cleavage and product formation by radical fragments coupling [18,33]. The biomass, composed of lignin, cellulose, and hemicellulose, had a number of covalent bonds, including $C_{al}$-$C_{al}$, $C_{ar}$-$C_{ar}$, $C_{ar}$-$C_{al}$, $C_{ar}$-O, $C_{al}$=$C_{al}$, $C_{al}$-O, and $C_{al}$=O bonds [34]. The cleavage of these bonds was correlated with the DTG behavior of the biomass. The TG and DTG curves of pine and the mixture are shown in Figure 2a,b.

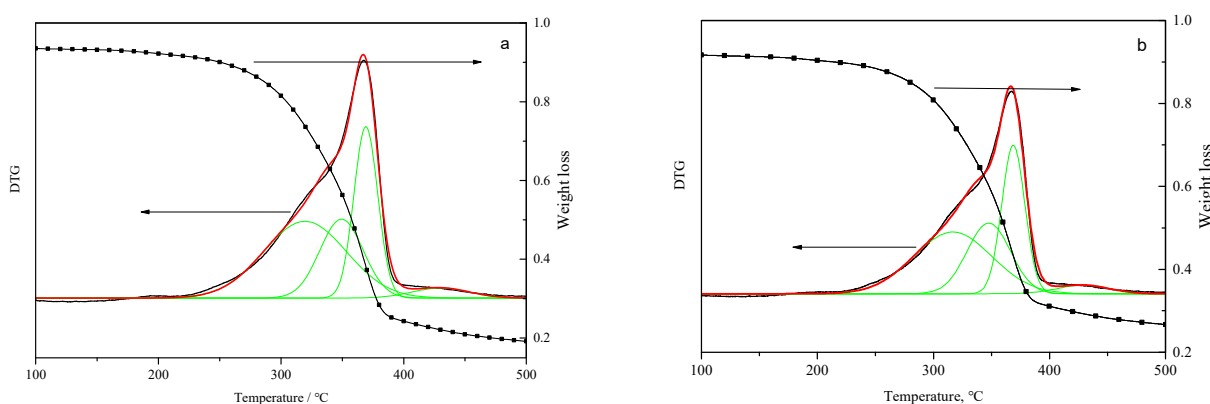

**Figure 2.** TG and DTG fitting curves between 100–500 °C of pine and mixture: (**a**) pine and (**b**) mixture with an aluminum dross to pine mass ratio of 1/4).

Figure 2 shows the cleavage of the covalent bond in the temperature range of 300–450 °C, which was mainly caused by the decomposition of carbonyl groups, the breakage of C-O-C, $C_{al}$-$C_{al}$, and side chains of aromatics during primary pyrolysis. The $C_{ar}$-$C_{ar}$, $C_{al}$=$C_{al}$, and $C_{al}$=O bonds could be not broken at a pyrolysis temperature between 300–450 °C. The DTG curves in Figure 2 show the overlapped signals for biomass pyrolysis. The signals were fitted with four sub-curves representing four major covalent bond cleavages [35]. Peak 1 is correlated to the decomposition of carbonyl groups, hydrogen bonds, weak C-O bonds, hemicellulose pyrolysis, and the disentanglement of cellulose, hemicellulose, and lignin in the biomass at about 320 °C [36]. The peak width is larger than the other peak. Peak 2 is dominated by cleavage of C-O bonds and weak $C_{al}$-$C_{al}$ bonds with a bond energy range of 295–325 kJ/mol [34] at about 350 °C. This pyrolysis stage was the decomposition of glucosidic linkage and amorphous cellulose. Peak 3 is correlated to the breakage of $C_{al}$-$C_{al}$ bonds at about 370 °C, associated with cellulose pyrolysis and weak aromatic side chain breakage. Peak 4 can be assigned to strong aromatic side chains and condensation of aromatic rings. The weight loss was the lowest at this terminal pyrolysis stage, as shown in Figure 2. The three parameters of the curve fitting operation, i.e., peak temperature ($T_p$), peak width ($W_p$) and peak area ($A_p$), calculated from Figure 2a,b are listed in Table 2.

The bond breakage behavior can be closely related to the peak temperature and peak area. The peak temperatures were similar with and without the aluminum dross, indicating little changes in the biomass pyrolysis pathway. Peak area was a quantitative description of the amount of bond cleaved at the respective temperatures. The peak area percentage of peak 1 decreased from 44.03% to 41.10% with AD addition, indicating that

the biomass pyrolysis rate was changed. The hemicellulose pyrolysis and macromolecular disentanglement were delayed. On the contrary, the peak area percentage of peak 2 increased from 23.69% to 25.95%, demonstrating that the cleavage of C-O bonds and weak C-C bonds as accelerated with AD addition. Pyrolysis of cellulose, especially amorphous cellulose, was activated by the catalytic effect of AD. Peak 3, associated with the breakage of C-C bonds, increased. Though the area percentages of peak 4 were similar, the residual content at the end of reaction was higher with AD addition, since the coking reaction during lignin pyrolysis was promoted.

**Table 2.** Fitting results of the DTG curve.

| Peak | DTG-Pine | | | DTG-Pine+AD | | |
|---|---|---|---|---|---|---|
| | $T_p$/°C | $W_p$/°C | $A_p$/% | $T_p$/°C | $W_p$/°C | $A_p$/% |
| 1 | 320 | 79 | 44.03 | 317 | 79 | 41.10 |
| 2 | 349 | 41 | 23.69 | 348 | 44 | 25.95 |
| 3 | 369 | 23 | 28.32 | 369 | 23 | 29.22 |
| 4 | 430 | 54 | 3.96 | 429 | 52 | 3.73 |

3.1.2. Effect of Aluminum Dross on Free Radical Concentration in Pyrolysis Residue

The free radical concentration in pyrolysis residue, char, formed with increasing temperature, was analyzed by EPR and is presented in Figure 3.

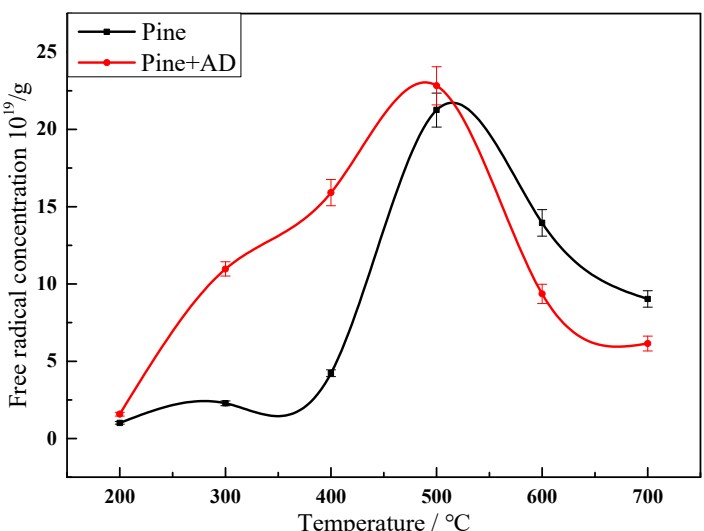

**Figure 3.** Variation on free radical concentrations in pyrolysis residue from pine and a mixture at different temperatures of 200–700 °C.

Biomass pyrolysis undergoes an activation process, generation of free radical fragments, and coupling of these radicals. The free radical concentration in biomass pyrolysis residue was about $10^{19}$ spins/g and increased to a maximum value of $10^{20}$ spins/g at 500 °C during pyrolysis. It is interesting to note that the temperature for maximum free radical concentration was higher than the terminal pyrolysis temperature. Free radical concentration decreased after 500 °C due to condensation reactions where the weight loss reached a plateau, as shown in Figure 2. During the co-pyrolysis of AD and pine, the biomass components, especially hemicellulose, were melted and actively combined by hydrogen bonds with AD at about 300–350 °C. Then, C-C bonds and C-H bonds were activated to form C-O-M (M is the metal) intermediates. The free radicals were trapped in the molten cluster [37]. The weight loss rate was delayed and the area percentage of peak 1 decreased, as shown in Figure 2. The free radical concentration in PADC increased from $2.28 \times 10^{19}$ to $10.97 \times 10^{19}$ spins/g at 300 °C, as illustrated in Figure 3. The maximum

free radical concentration of $22.82 \times 10^{19}$ spins/g was reached at 500 °C, correlating to the cleavage of C-O and C-C bonds in C-O-M intermediates; the value was similar to that in pure pine pyrolysis residue. It demonstrates that AD accelerates the free radical generation during pine pyrolysis. The results agree well with the DTG behavior in Figure 2 and Table 2. The areas of peaks 2 and 3 increased with AD addition, demonstrating faster reaction. The free radical coupling reaction was also promoted by AD, indicated by the steeper decline in free radical concentration, as shown in Figure 3.

### 3.2. Correlation of AD Catalytic Effect to Structural Evolution during Pyrolysis

The primary pyrolysis was terminated at 430 °C, as illustrated in DTG curves, whereas the free radical concentration increased to the maximum value at 500 °C. The carbon structure of the pyrolysis residue from pine and the mixture at 500 and 700 °C were analyzed by $^{13}$C-NMR shown in Figure 4a–e.

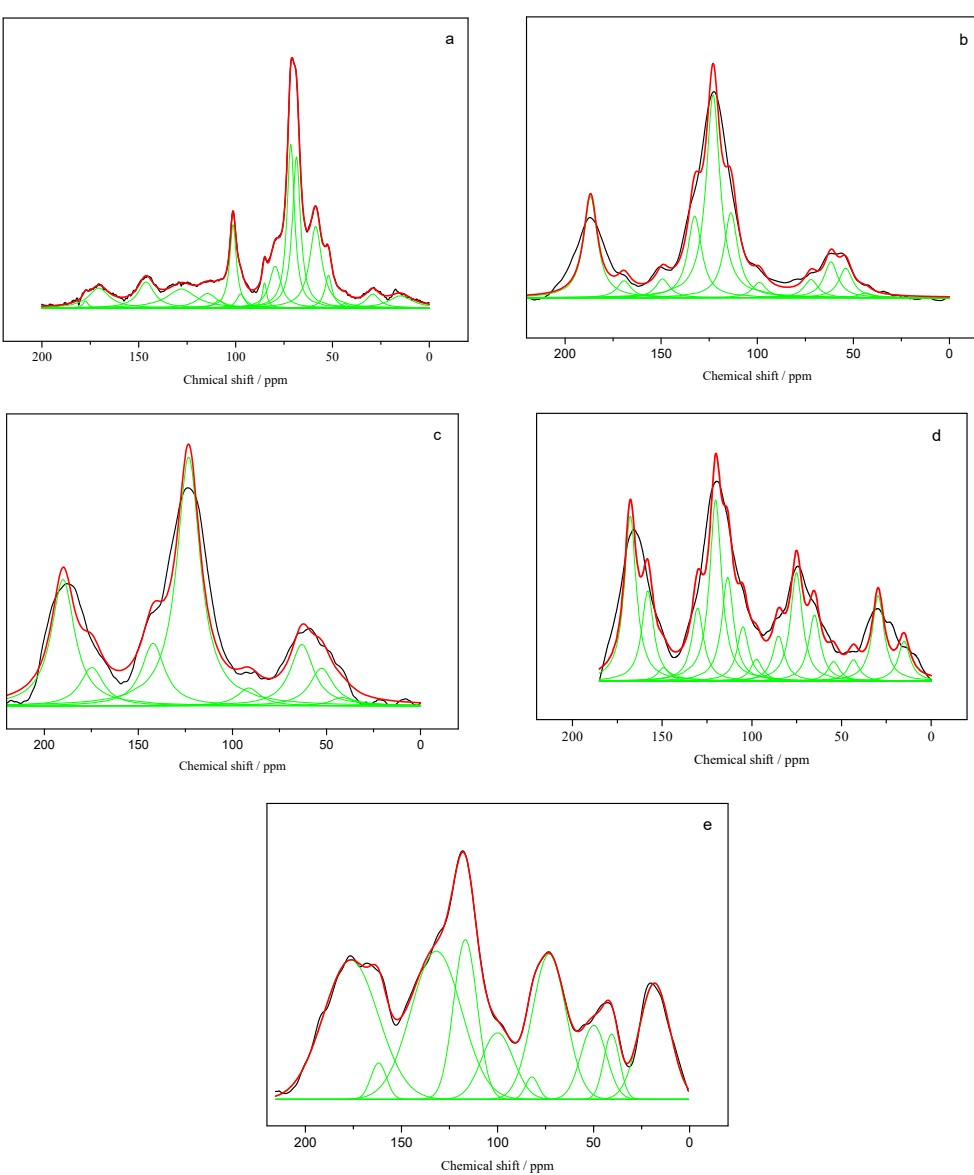

**Figure 4.** Fitting $^{13}$C-NMR curves of samples: (**a**) pine, (**b**) PC-500, (**c**) PC-700, (**d**) PADC-500, and (**e**) PADC-700.

Figure 4a–e shows the curve fitting results of char from pine pyrolysis and co-pyrolysis. The biomass spectrum in Figure 4a has relatively sharp signals at 15–110 ppm due to carbohydrate polymers from the cellulose and hemicellulose, which were facilely broken

during primary pyrolysis before 500 °C. With pyrolysis temperature increasing, signals at 110–170 ppm corresponding to the aromatic structure became protuberated and the types of carbon corresponding to the number of sub-curves were fewer. The region at 170–220 ppm contributed by carbonyl groups had a higher intensity in char due to the conjugation of the oxygen functional group to an aromatic structure. When AD was added to pine for co-pyrolysis, the signals at 50–100 ppm were sharp in comparison to char from pine pyrolysis. The carbon types with the oxygen functional group in PADC were more abundant than that in PC. The catalytic pyrolysis with AD generated amounts of active intermediates.

### 3.2.1. Aliphatic Structure Activation by AD

Comparing the sub-curves and area percentage assigned to the aliphatic structures in PADC with those in PC, PADC had a higher area percentage at 15, 30, 72, 80, and 101 ppm, as shown in Figure 5. The pyrolysis mechanism was summarized using biomass models by other researchers [38,39]. The initial step for catalytic pyrolysis by metallic oxide was the formation of active intermediates, with the oxidized clusters of C-O-M following the intermediate hybrid mechanism [40–42]. The oxygen functional groups were more easily activated by metallic oxide [40,41]. AD and pine melt mixed well with each other to accelerate mass transfer and generate active radicals. The oxygen-containing functional group of the aliphatic structure in biomass components, such as alcohol hydroxyl group and ether, could form C-O-M clusters of C-O-M with the metallic oxide in AD. The area percentage of the signal at 72 ppm assigned to C2,3,5 from hemicellulose or cellulose in char, which linked with the alcohol hydroxyl group, increased from 3.00% to 10.13% with AD addition at 500 °C. The signal at 15 ppm assigned to $CH_3$-COO- from hemicelluloses increased from 0 to 3.49%. When the temperature increased to 700 °C, these oxygenated structures were still present in char due to the strong conjugation to the aromatic structure and the percentage increased. These structures contributed to the formation of the aggregated structure in char. This was evidenced by the increase in pyrolysis residue weight with AD addition, as shown in Figure 2. More glycosidic bonds linked with oxygen functional groups were activated by AD during pyrolysis and trapped in molten clusters to pyrolysis. At 500 °C, the correlating signal for C4 in amorphous cellulose at 80 ppm increased from 0 to 4.21% and that for C1 in hemicellulose at 101 ppm increased from 0.02% to 5.07%. These active intermediate structures were then facilely broken with further increase in temperature and were no longer detected in the composite at 700 °C. Meanwhile, the bridging aliphatic structure ($CH_2$) corresponding to the signal at 30 ppm was also trapped in the intermediate structure during primary pyrolysis; an area percentage of 7.78% was observed in PADC-500. This structure disappeared with higher temperature and the area percentage became 0 in PADC-700. However, C1 in cellulose assigning at 109 ppm, correlating to the crystalline structure, decreased from 14.00% to 9.66% due to catalytic pyrolysis. This was in accordance with the slightly larger peak areas for peaks 2 and 3 in the DTG sub-curves. The structural evolution indicated that the AD had a significant influence on the pine pyrolysis rate, favoring the formation of active radicals containing an alcohol hydroxyl group and glycosidic bonds.

### 3.2.2. Aromatic Structure Activation by AD

Aromatic oxygenated carbon corresponded to signals at 146 and 164 ppm in [13]C-NMR spectra. The area percentage of the signal at 164 ppm assigning to etherified syringyl aromatic carbon in PADC-500 was higher than that in PC-500, as shown in Figure 6; it was then further activated and decreased to 1.71% in PADC-700. However, the etherified syringyl aromatic carbon from pure pine pyrolysis increased with temperature. Similarly, the etherified guaiacyl aromatic carbon assigning at 146 ppm in PC-500 and PC-700 was also higher than that in PADC-500 and PADC-700, respectively, mostly due to the catalytic activation. The increase of bridging aromatic carbon assigning at 129 ppm and the decrease of proton aromatic carbon assigning at 116 ppm also suggested that AD catalyzed pyrolysis

had enhanced mass transfer in melted composite and the formation of side chain substituted aromatic rings. Catalytic pyrolysis produced a C-O-M intermediate to promote the decomposition of the oxygenated aromatic structure.

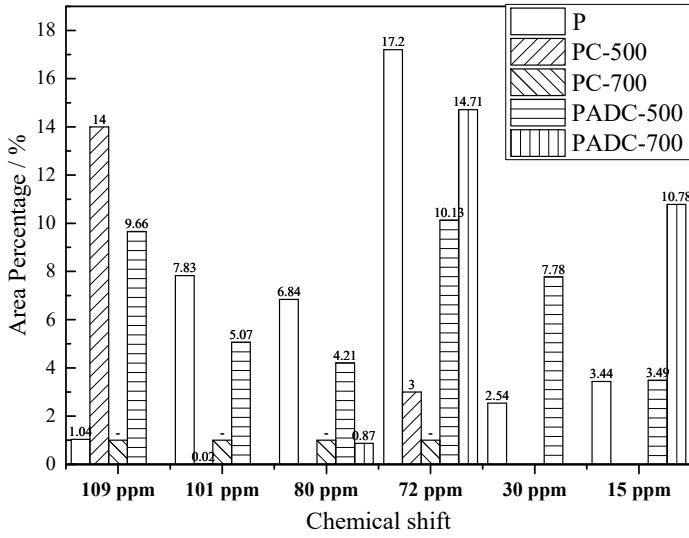

**Figure 5.** Variation of the aliphatic carbon structure area percentage assigned at chemical shift.

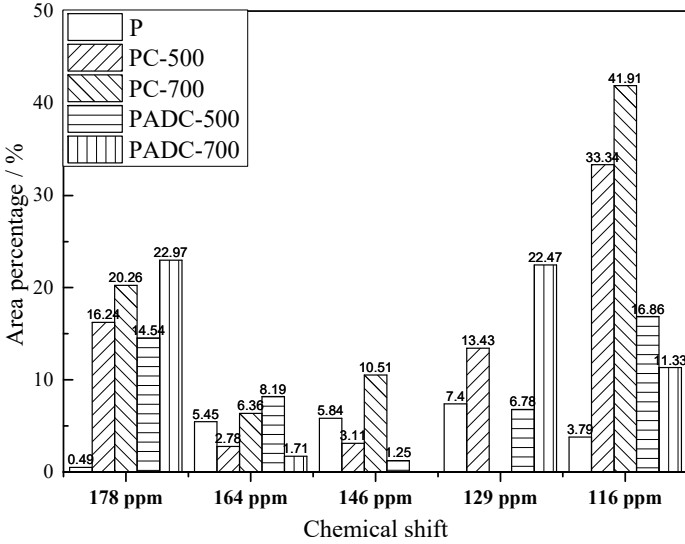

**Figure 6.** Variation of the aromatic carbon structure area percentage assigned at chemical shift.

### 3.3. Proposed Catalytic Pyrolysis Mechanism for Tar Formation

As discussed above, AD had an important influence on the pine pyrolysis rate and contributed little to the thermo-chemical equilibrium. Figure 7 shows the catalytic pyrolysis pathway. The three main components, cellulose, hemicellulose, and lignin in pine, melted with AD first. The functional groups, such as hydroxyl, carbonyl, and ether, combined with metallic oxide in AD to form C-O-M clusters by hydrogen bond or van der Waals forces. The number of radical fragments trapped in residue increased rapidly to $10.97 \times 10^{19}$ spins/g at 300 °C, as shown in Figure 3. The decomposition of functional groups was suppressed so that the peak 1 area in the sub-curve of DTG decreased. The cleavage of ether linkage in cellulose, such as C1, was accelerated by AD, leading to an increment in the peak 2 area in the DTG sub-curves. Moreover, oxygenated aromatic carbons were activated by AD to break facilely at higher temperature. As the C-O-M clusters aided to form a macromolecular network and hence aggregate structures by a conjugation effect, a number of carbonyl and hydroxyl groups remained in the PADC residue.

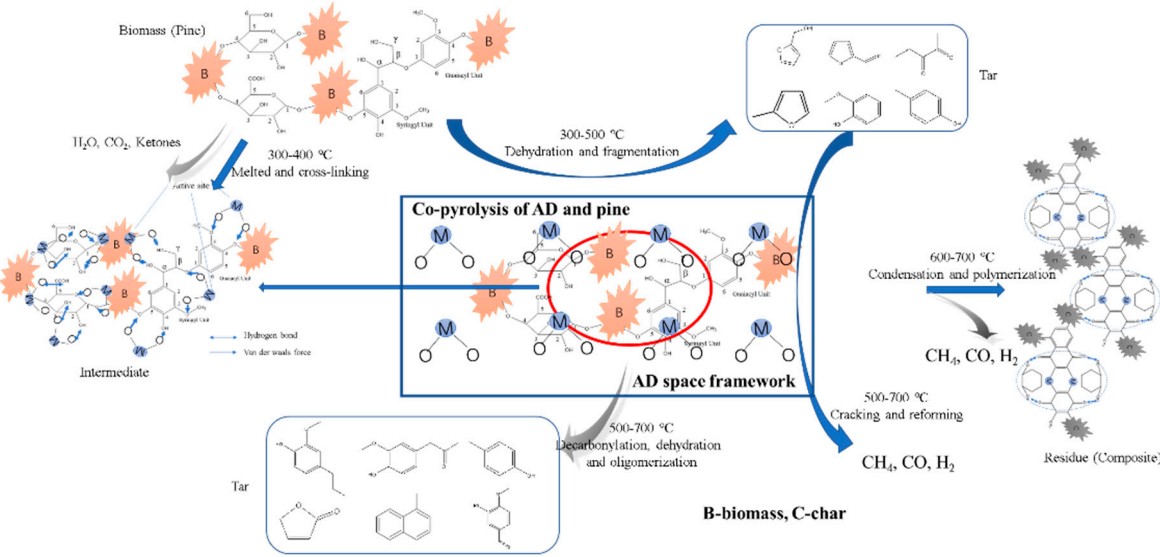

**Figure 7.** Proposed reaction pathway of catalytic pyrolysis on AD.

Tar compositions from fixed bed pyrolysis at 500 and 700 °C were analyzed by GC-MS and are shown in Figure 8.

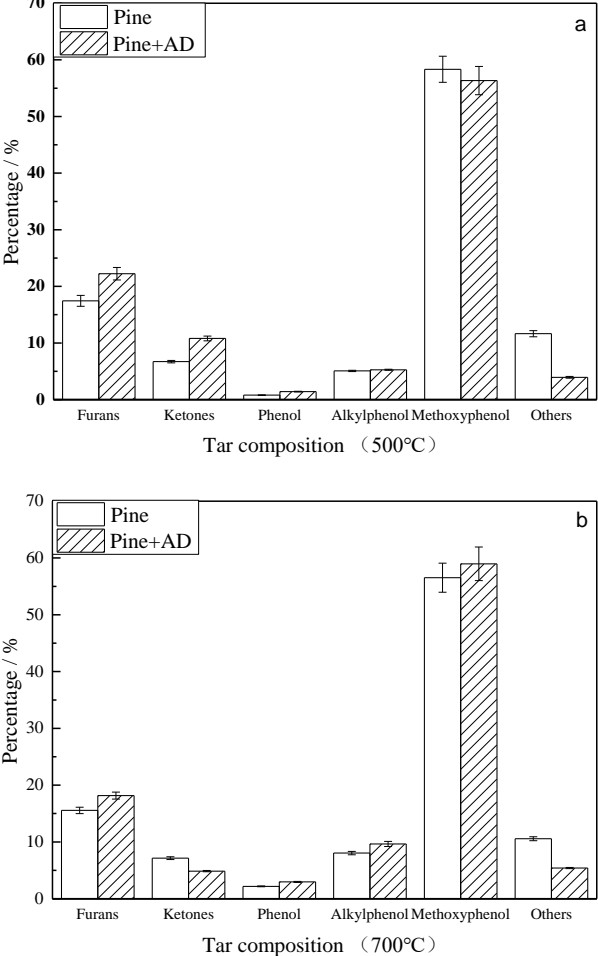

**Figure 8.** Comparison of the tar composition from the pyrolysis of pine and a mixture at 500 °C and 700 °C: (**a**) 500 °C and (**b**) 700 °C.

The primary tar, mainly including furans, ketones, phenol, alkylphenols, and methoxyphenols, was formed by a free radical reaction. Furans and ketones were formed from cellulose and hemicellulose by the cleavage of C-O bonds. The catalytic pyrolysis accelerated the breakage of C-O bonds connecting C1 (shown in Figure 7) in cellulose to form ketones by scheme 1 in Figure 9 [12]. Part of the glycosidic bonds were cleaved after activation of C1 in hemicellulose by AD. A substantial amount of active intermediate the radicals further decomposed into active monomers containing hydroxyl radicals and macromolecular radicals. C-C bonds in active monomers broke down to form smaller radicals, and then ketones. The ketones from catalytic pyrolysis were higher than those from non-catalytic pyrolysis at 500 °C, suggesting that the primary pyrolysis for ketone formation via scheme 1 was accelerated. This was in agreement with the larger area percentage of peak 2 in the DTG sub-curves of catalytic pyrolysis. With a pyrolysis temperature of 700 °C, the amount of ketones from pure pine pyrolysis changed insignificantly. However, the ketones from catalytic pyrolysis decreased from 10.80% to 4.86% as a result of secondary pyrolysis to generate gas on AD.

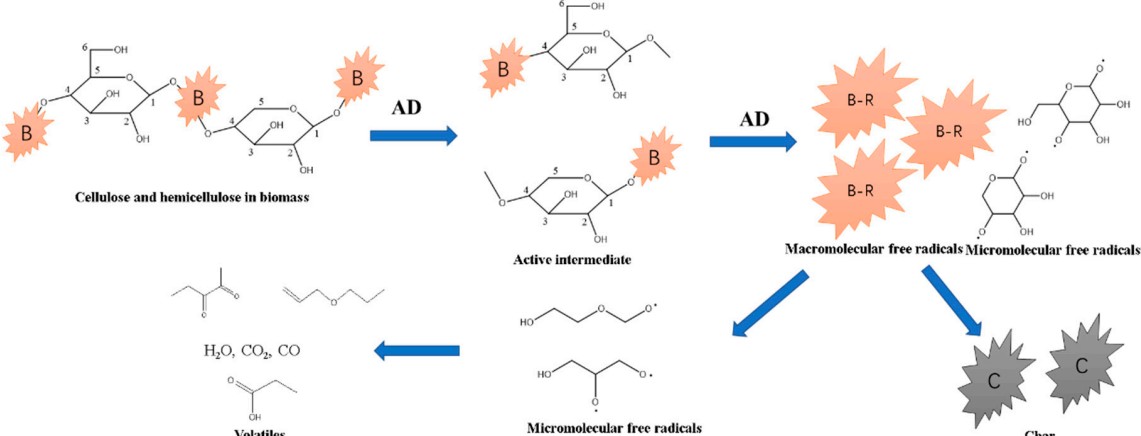

**Figure 9.** Scheme 1 for formation of ketones from cellulose and hemicellulose catalyzed by AD.

Figure 10 shows the formation scheme of furans. It was first initiated by the thermal cleavage of the C-O bond in the hexatomic ring of a cellulose or hemicellulose monomer [12,13]. Then, the dihydroxylation and transformation of the structure configuration were induced to form furan rings (5-hydroxy methyl furfural, HMF). HMF decomposed to form furans. As discussed in Figure 5, AD enhanced the cleavage of the C-O bond in C1 of cellulose, forming selective furans and ketones. The percentage of furans was 22.23% from catalytic pyrolysis, higher than 17.45% of pure pine pyrolysis at 500 °C.

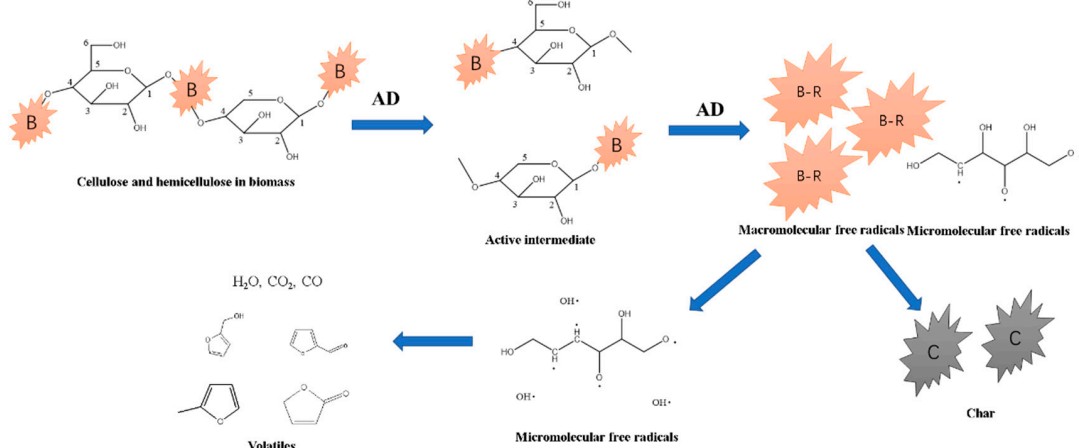

**Figure 10.** Scheme 2 for the formation of furans from cellulose and hemicellulose catalyzed by AD.

Phenols were the major components in primary bio-tar, formed mainly via lignin pyrolysis, as illustrated by Scheme 3 in Figure 11 [15–17]. The amount of methoxyphenols produced from catalytic pyrolysis at 500 °C was lower than that from non-catalytic pyrolysis; however, at 700 °C, the catalytic phenols via co-pyrolysis increased from 63.03% to 71.57%, higher than the non-catalytically produced phenols. Some glycosidic bonds were trapped in the molten clusters with AD, as discussed in previous sections. The pyrolysis intermediates of anhydro-oligosaccharides could have produced phenols via the catalytic pyrolysis, as shown in scheme 4 in Figure 12 [43], leading to an increment of phenols at 700 °C. Meanwhile, the etherified guaiacyl assigning at 146 ppm was more active to be ruptured and formed, contributing to phenol formation at 700 °C. Both factors resulted in higher phenol production via co-pyrolysis at elevated temperatures.

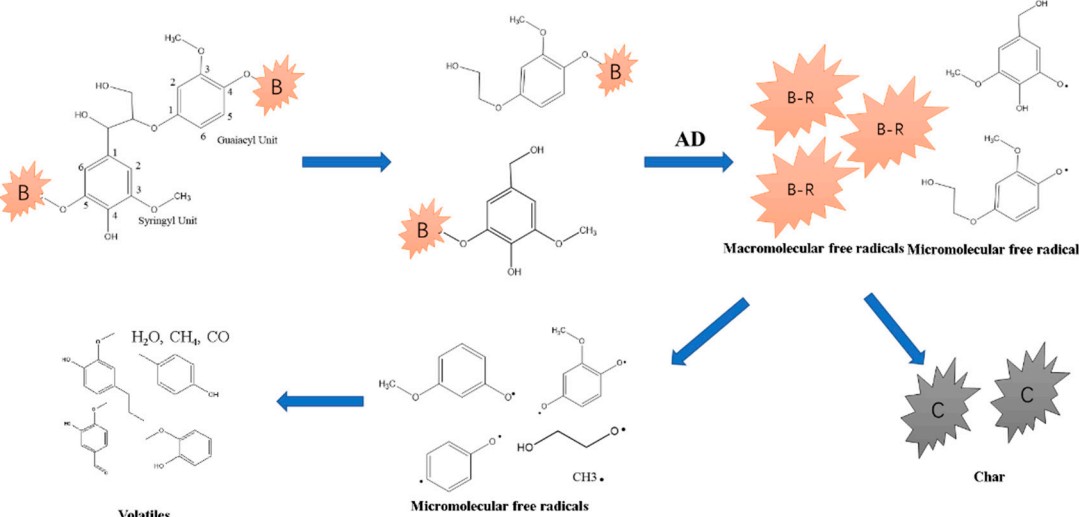

**Figure 11.** Scheme 3 for the formation of phenols from lignin catalyzed by AD.

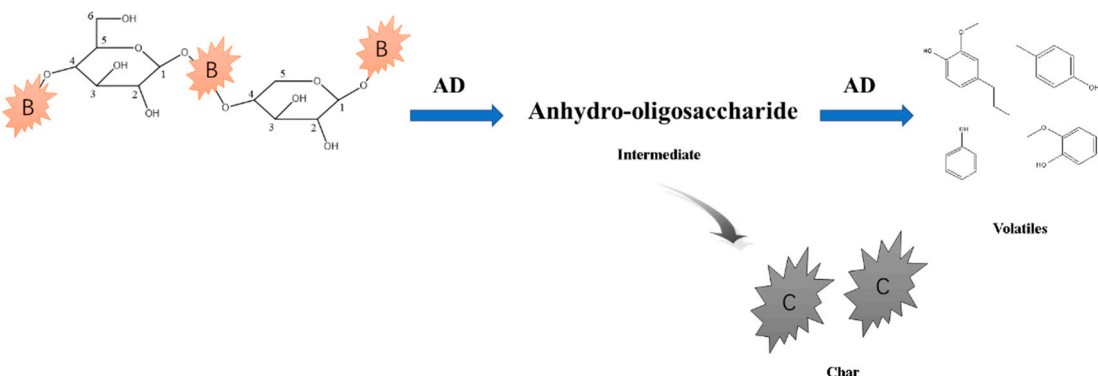

**Figure 12.** Scheme 4 for the formation of phenols from cellulose and hemicellulose catalyzed by AD.

The content of components other than furans, ketones, and phenols in tar was much lower from catalytic pyrolysis. AD improved selectivity for furans, ketones, and phenols, which were higher value-added products [44]. Further, the catalytically produced tar could be purified at facile conditions via cracking and reforming in the successive processes [44]. The energy demand was decreased and the overall economic benefit could be promoted.

## 4. Conclusions

The primary tar formation mechanism via catalytic pyrolysis on AD was analyzed. Some conclusions are summarized as follows:

(1) AD accelerates the cleavage of C-O bonds to form a great amount of free radicals, which are confined in the aggregated structure of AD and biomass during pyrolysis.

(2) Catalytic pyrolysis on AD to produce char, tar, and gas is initiated by the formation of active C-O-M intermediates. The broken C-O bonds linked to etherified guaiacyl and C1 of celluloses are accelerated; the glycosidic bonds linking with oxygen functional groups are activated to cleave at 700 °C. The alcohol hydroxyl and carbonyl groups remain in the residue to form aggregated structures in char.

(3) AD enhances the selective formation of primary tar components such as ketones, furans, and phenols. It accelerates the cleavage of C-O bonds linked with C1 of celluloses and hemicellulose to form ketones and furans. Moreover, AD promotes the formation of phenols via the rupture of the etherified guaiacyl bond and anhydro-oligosaccharide at high temperatures.

These results may facilitate the process optimization and reactor design of commercial-scale biomass gasification and provide a new direction for the reuse of organic and industrial wastes. The goal of simultaneous energy regeneration from biomass and comprehensive utilization of industrial waste may be achieved.

**Author Contributions:** Conceptualization, P.L. and Z.Z.; formal analysis, L.L.; investigation, L.L. and P.L.; resources, P.L.; data curation, L.L. and Q.W.; writing—original draft preparation, P.L.; writing—review and editing, Z.Z.; visualization, H.Y.; supervision, H.T. and T.Z.; funding acquisition, H.Y. and H.T. All authors have read and agreed to the published version of the manuscript.

**Funding:** This study was funded by the Natural Science Foundation of China (51906021 and 51703014), National Key R&D Program of China (2018YFC1901203), Natural Science Foundation of Colleges in Jiangsu Province (19KJB480005), Changzhou applied basic research plan (CJ20190081), Research Foundation for Advanced Talents of and Changzhou University (ZMF17020034), and Mid Long-Term Development Strategy research on Engineering Technology of China (2019ZCQ04).

**Institutional Review Board Statement:** Not applicable for studies not involving humans or animals.

**Informed Consent Statement:** Not applicable for studies not involving humans.

**Data Availability Statement:** It does not concern the contents of data records deposited by the platform's users. However, all these data records are made available in the spirit of the idea of open data. The platform includes both resources that can be used without restrictions (public domain), and those that are shared under certain conditions (e.g., author's attribution, no modification). They can be searched, viewed, and downloaded. They can be printed, cited, and redistributed for the most part, as long as the use complies with the licenses' terms.

**Conflicts of Interest:** The authors declare no conflict of interest.

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
