# Peer review of "Insight into Tar Formation Mechanism during Catalytic Pyrolysis of Biomass over Waste Aluminum Dross"

_applsci, doi:10.3390/app11010246_

Round 1

Reviewer 1 Report

This is a manuscript on an important topic in gasification and pyrolysis. Although there are many articles on tar formation mechanisms already, this manuscript forms a valuable addition. It reports on very thorough analysis work, the results are well discussed in a detailed, scientifically sound way and the writing is concise.

However, the materials & methods should be described in a more elaborate way. This may comprise:

  • Include a figure of the experimental pyrolysis set-up
  • Provide details of the pyrolysis reactor (e.g., diameter) and the connected gas-liquid separator
  • Provide more details of the sawdust, e.g. particle diameter (distribution)
  • Provide more details of the aluminium dross, e.g. particle diameter (distribution), morphology, (if available) distribution of elements/chemical speciation

Moreover, a critical check of the English language is recommended (e.g., singular/plural).

Author Response

Issue 1: Include a figure of the experimental pyrolysis set-up

Discussion 1: We have added pyrolysis set-up in manuscript as Fig.1.

Issue 2: Provide details of the pyrolysis reactor (e.g., diameter) and the connected gas-liquid separator

Discussion 2: We have provide details of pyrolysis reactor as “Pyrolysis tests were carried out in a vertical quartz tubular reactor. The reactor was made of quartz tube with an external diameter of 40 mm and an overall length of 750 mm. The gas-liquid separator was connected with the reactor through ground glass, which is shown in Fig 1” in manuscript.

Issue 3: Provide more details of the sawdust, e.g. particle diameter (distribution)

Discussion 3: We have provided details of the sawdust as “Pine sawdust is collected from south of Anhui province in China, were used as the raw material. They were ground to <0.5mm and stored under cryogenic environment for subsequent analysis” in manuscript.

Issue 4: Provide more details of the aluminum dross, e.g. particle diameter (distribution), morphology, (if available) distribution of elements/chemical speciation

Moreover, a critical check of the English language is recommended (e.g., singular/plural).

Discussion 4: We have provided details of aluminum dross as “AD is produced from the secondary smelters during the dross recycling process of a factory in Changzhou, which is identified as the hazardous wastes to be disposed and rich in Al2O3 as catalyst for pyrolysis. The main chemical composition of AD analyzed by X-ray fluorescence spectrometer (XRF, S8 Tiger, Bruker AXS, Germany) is 78.38% of Al2O3, 7.77% of SiO2, 5.18% of Na2O, 4.51% of MgO, 0.68% of CuO, 0.74% of Fe2O3, 0.51% of K2O, 0.39% CaO and 0.19% of ZnO. They were ground to <0.1mm for catalytic pyrolysisin manuscript.

Reviewer 2 Report

The paper is interesting and scientifically important. It can be published after corrections: 1) 2. Materials and Methods 2.1. Materials: Pine sawdust and their mixtures (please describe precisely, why this mixtures composition was chosen, for which research results are presented in: Results and discussion) 2.2. Methods 2.2.1. Pyrolysis 2.2.2. Mass change of sample by TGA 2.2.3. Composition analysis of pyrolysis tar by GC-MS 2.2.4. Characterization of carbon structure and free radical concentration of solid product from pyrolysis 2) Please describe lines in Figure 1a, b, and Figure 3 (insert legend in graphs). 3) How many times were measurements repeated (measurements of each type, presented in the paper)? Please insert Figures showing the repeatability of measurements. Please calculate errors of all measurements, presented in the paper.

Author Response

Issue 1: 2. Materials and Methods 2.1. Materials: Pine sawdust and their mixtures (please describe precisely, why this mixtures composition was chosen, for which research results are presented in: Results and discussion) 2.2. Methods 2.2.1. Pyrolysis 2.2.2. Mass change of sample by TGA 2.2.3. Composition analysis of pyrolysis tar by GC-MS 2.2.4. Characterization of carbon structure and free radical concentration of solid product from pyrolysis.

Discussion 1: We have re-edited the structure of materials and methods in revised manuscript as reviewer suggested.

Issue 2: Please describe lines in Figure 1a, b, and Figure 3 (insert legend in graphs).

Discussion 2: We have described the lines in subsequent manuscript detailly and given the illustration in the manuscript. (Table 2 for Fig 2a-b, Fig 5-6 for Fig 4a-e).

Issue 3: How many times were measurements repeated (measurements of each type, presented in the paper)? Please insert Figures showing the repeatability of measurements. Please calculate errors of all measurements, presented in the paper.

Discussion 3: All measurements in manuscript are repeated for more than three times. We have insert Fig 4 and 9 showing errors of measurements.